# GaN JBS Diode Device Performance Prediction Method Based on Neural Network

**DOI:** 10.3390/mi14010188

**Published:** 2023-01-12

**Authors:** Hao Ma, Xiaoling Duan, Shulong Wang, Shijie Liu, Jincheng Zhang, Yue Hao

**Affiliations:** School of Microelectronics, Xidian University, Xi’an 710071, China

**Keywords:** GaN JBS diode, TCAD simulation, neural network, performance prediction

## Abstract

GaN JBS diodes exhibit excellent performance in power electronics. However, device performance is affected by multiple parameters of the P+ region, and the traditional TCAD simulation method is complex and time-consuming. In this study, we used a neural network machine learning method to predict the performance of a GaN JBS diode. First, 3018 groups of sample data composed of device structure and performance parameters were obtained using TCAD tools. The data were then input into the established neural network for training, which could quickly predict the device performance. The final prediction results show that the mean relative errors of the on-state resistance and reverse breakdown voltage are 0.048 and 0.028, respectively. The predicted value has an excellent fitting effect. This method can quickly design GaN JBS diodes with target performance and accelerate research on GaN JBS diode performance prediction.

## 1. Introduction

The vertical GaN Schottky diodes have been widely used in high-power electronic circuits due to their low switching voltage and fast switching performance [1]. However, the performance of the vertical GaN Schottky diode still has many shortcomings, such as the reverse breakdown voltage is not ideal. The GaN junction barrier Schottky diode (JBS) combines the advantages of both the Schottky diode and PIN diode. It has low on-state resistance and high reverse breakdown voltage, which can significantly improve the performance of power electronics systems [2].

However, the on-state resistance and breakdown voltage of the GaN JBS diode is affected by multiple parameters in the P+ region [3]. The traditional device simulation and experimental test methods have a long cycle and low efficiency, so design requires a lot of human resources. The rapid development of neural network provides another choice for rapidly predicting the structure or properties of devices and materials. There is research on the performance prediction of neural networks on MOSFET and SiC devices and GaN materials [4,5,6]. A GaN JBS diode device has more influence parameters and a more complex mechanism, and a more complex neural network model is necessary to describe the device accurately. Therefore, the neural network structure needs to be fully optimized.

This paper proposes a method to predict and optimize the performance of GaN JBS diodes using an optimized neural network. The network input determined by us includes the doping concentration in the drift region (Epidop), the doping concentration in the P+ region (Impdop), the ratio of the width of the P+ region to the spacing between adjacent P+ regions (L), and the injection depth of the P+ region (Impthickness). The network output includes the on-state resistance (Ron) and reversed breakdown voltage (BV). Then, TCAD Sentaurus was used to accumulate the sample data. After training the data, accuracy and mean relative error (MRE) were used to evaluate the prediction results.

## 2. GaN JBS Diode TCAD Modeling and Simulation

### 2.1. Device Structure Simulation

This paper’s device simulation and sample data accumulation are based on TCAD Sentaurus. Figure 1a shows the device model of GaN JBS diode. The anode is defined as a Schottky contact, and the cathode is defined as an ohmic contact. The JBS structure is formed on the device surface in four gauss-doped P-type regions. The drift region and substrate are N-doped. When the reverse voltage is applied to the device, the PN junction composed of P+ region and N-type drift region can withstand voltage. The large electric field falls in the P+ region, thus increasing the breakdown voltage. The specific structural parameters of the device are shown in Table 1 [7].

Among them, parameter L was set as half of the width of P+ region to present the width of P+ region and the spacing between adjacent P+ regions.

TCAD Sentaurus analyzed the forward and reverse characteristics. Some critical physical models were applied, including avalanche ionization, high field mobility, bandgap narrowing, doping dependence, and Auger recombination. The electrode voltage signal was set; the forward and reverse IV curves are shown in Figure 1b,c. We can see that on-state resistance (Ron) is 0.938 mohm, and breakdown voltage (BV) is 549 V. The simulation results show that the GaN JBS diode model based on TCAD Sentaurus accords with reality.

### 2.2. Data Gathering

The P+ region and the drift region greatly influence the forward and reverse characteristics of the GaN JBS diode. To better represent the device, the input parameters were determined as doping concentration of drift region (Epidop), doping concentration of P+ region (Impdop), a ratio of the width of P+ region to the spacing between adjacent P+ region (L), and injection depth of P+ region (Impthickness). To obtain accurate prediction results, it is necessary to set the input parameters reasonably. Figure 2 shows the influence of input parameters on Ron and BV based on TCAD Sentaurus simulation. Through simulation results, the reasonable range of Epidop, Impdop, L, and Impthickness should be 3 × 10^15^–1 × 10^16^ cm^−3^, 3 × 10^17^–1 × 10^18^ cm^−3^, 0.2–0.6 µm, 0.15–0.4 µm [8]. Table 2 lists the specific values of each input parameter.

Then, the TCAD Sentaurus tool was used to conduct a simulation according to the values of the above variables, and extract Ron and BV corresponding to each sample from the results. After removing the samples that failed in the simulation, a data set with a sample capacity of 3018 was finally formed (Appendix A). Datasets with sufficient samples can effectively improve the generalization of the model. Then, 3018 sets of datasets were divided into the training set, verification set, and test set at the proportion of 8:1:1. The input and output data were standardized and normalized in the training set and verification set [5]. To truly reflect the neural network’s generalization, the algorithm should not know the information about any test set, so the variance and mean of the test set came from the primary data of the training set.

## 3. Establishment of Neural Network Structure

To better extract data features, this paper uses a convolutional neural network [9]. Neural network architecture is composed of an input layer, hidden layer, and output layer [10]. The forecasting object determines the input and output layers. Figure 3a shows the basic network structure, and the key to design lies in the hidden layer. This paper’s hidden layer includes all connection modules, convolution modules, and batch-normalized layers. Each layer of the network structure is described below.

Input and output layer. According to the established data set, the doping concentration of drift region (Epidop), the spacing of P+ region (L), the injection depth of P+ region (Impthickness), and the injection concentration of P+ region (Impdop) were set as inputs. On-state resistance (Ron) and breakdown voltage (BV) were set as outputs. Therefore, the network architecture has four inputs and two outputs.Fully connected module. Since the dimension of the ground input vector of the dataset is small, a fully connected module [11] was added after the input layer for dimension expansion to facilitate subsequent convolution operations. In addition, a batch normalization layer [12] was added after each complete connection layer to prevent overfitting.Convolution module. The convolution layer of neural network architecture established in this paper includes a transposed convolution module, a double-branch convolution module, and a convolution module. Unlike the convolution module, the transposed convolution module [13] can expand the data dimension. Therefore, the transposed convolution module was added to expand the input dimension further. The dual-branch convolution module can extract data features and prevent gradient disappearance or explosion. The structure of the double-branch convolution module is shown in Figure 3b. The output features of the two channels were then spliced, and the spliced features were used as the output of the dual-branch convolution module. In addition, a batch normalization layer was added between each convolutional layer to prevent overfitting.

The above three parts constitute the network structure established in this paper. Meanwhile, the number of layers of each module in the hidden layer significantly impacts prediction results, so further optimization is needed. Figure 4 shows the process for determining the number of layers. Using the exhaustive method, multiple neural network structures with different layers were defined, and the data were input for training. The errors between the predicted and actual values were compared; the one with the slightest error was selected as the optimal neural network structure. The final determined network structure is shown in Figure 5. It consists of three input layers, two transposed convolutional modules, four double-branch convolutional modules, three convolutional modules, and three output layers. The number of neurons in each layer is also marked below the structure of each layer.

## 4. Predicted Results

In this paper, the Pytorch deep learning framework is rewritten in Python based on torch to implement acceleration specifically for GPU [14]. The framework is easy to use and supports dynamic computing graphs and efficient memory use. In addition, simulation prediction is carried out on the calculator platform based on RTX3060 and R7-5800H. Firstly, each training batch’s loss function, ReLu, was calculated [15]. Then, the ADAM [16] optimizer was used to backpropagate the network parameters until the convolutional neural network converges. At this point, we obtained the trained model. The prediction model uses the early stop [17] method to control whether the training is over. When the prediction error of the model on the verification set is not reduced or reaches a certain number of iterations, the training breaks, and the parameters in the previous iteration results are used as the final parameters of the model. The last saved network weight parameters are taken as the final model parameters. After training, mean relative error (MRE) was used to characterize the prediction effect. It is defined as
(1)$$\mathrm{MRE}=\frac{1}{N}\overset{N-1}{\underset{i=0}{\sum }}\frac{\left|y_{i}-f_{i}\right|}{y_{i}}$$
where $y_{i}$ and $f_{i}$ represent the predicted value and the true value, respectively.

Using a determined neural network model to train sample data, the predicted results (Appendix A) show that the mean relative errors of Ron and BV are 0.028 and 0.048, respectively. Figure 6 shows the Ron and BV comparison of the predicted and real values. For a clearer view of the predicted results, test group data were arranged from smallest to largest, and the corresponding predicted value changed accordingly.

Figure 6 shows the training loss in the training process; (a) and (b) are the training loss of Ron and BV, respectively. The loss in the training process gradually decreases and finally becomes stable. In Figure 7, the black symbol represents the true value, and the nearest red symbol represents the corresponding predicted value. The closer the red and black marks are, the better the predicted result is. There is an excellent fitting result between the predicted and real values. In addition, the fitting degree of Ron is higher than that of BV. In Figure 7d, there is a slight deviation between the predicted values and the real values of BV in the 220–300 V range. This phenomenon can be analyzed. Breakdown voltage (BV) is the voltage applied to the device when the reverse leakage current reaches 1 × 10^−6^ A/cm^3^. However, the leakage current may not reach this standard for different device structure parameters when the device is broken down. In this case, the breakdown voltage (BV) is determined by the maximum electric field. The kind of data also truly reflects the performance change of GaN JBS diode (Appendix A); these are not invalid data. Because of the two methods of collecting BV, the error of prediction result is relatively large.

To explain the predicted result more intuitively, a bar chart of relative errors is shown in Figure 8. The number of predicted values with minor mean relative errors is much more significant than that with large mean relative errors. Figure 8a shows that 90% of Ron’s prediction errors are in the range of 0 to 0.06, and the errors are all less than 0.08. In Figure 8b, 90% of BV’s prediction errors are in the range of 0 to 0.09, and the errors are all less than 0.11. All these prove that the prediction results are relatively ideal.

In addition, decision tree [18], K-nearest neighbor (KNN) [19], and support vector machine (SVM) [20] were used to train and predict text data samples and compare them with the predicted results of the convolutional neural network. Figure 9 shows the MRE of the predicted effects of three traditional machine learning methods and convolutional neural networks. In Figure 9a, the MREs of the decision tree, KNN, SVM, and DNN prediction results for Ron are 0.36211, 0.05868, 0.45824, and 0.028, in order. The MREs of the decision tree, KNN, SVM, DNN prediction results for BV are 0.25708, 0.13592, 0.53633, 0.048 in order. The neural network structure established in this paper has obvious advantages in predicting the performance of GaN JBS diodes.

## 5. Conclusions

This paper applies the neural network to the performance prediction of GaN JBS diode devices. The input parameters are adjusted according to the prediction results to optimize the device characteristics. A total of 3018 groups of sample data, including device structure and performance parameters, were obtained by TCAD tool simulation, and the neural network was used for training. The results show a superb fitting result between the predicted and real values. The mean relative error of on-state resistance and reverse breakdown voltage is only 0.028 and 0.048, respectively. It can quickly and conveniently establish the association between GaN JBS diode device structure and performance index, accelerating the research of GaN JBS diode performance prediction.

## Figures and Tables

**Figure 1 micromachines-14-00188-f001:**
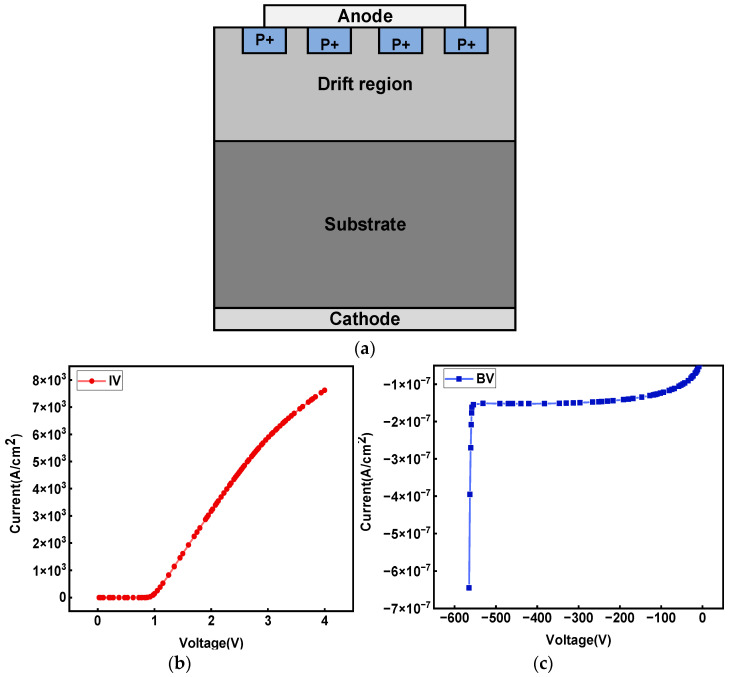
Figure (**a**) shows the schematic diagram of GaN JBS device structure, and the forward and reverse features of GaN JBS device under the structural parameters given in Table 1: (**b**) forward I-V curve, (**c**) reverse I-V curve.

**Figure 2 micromachines-14-00188-f002:**
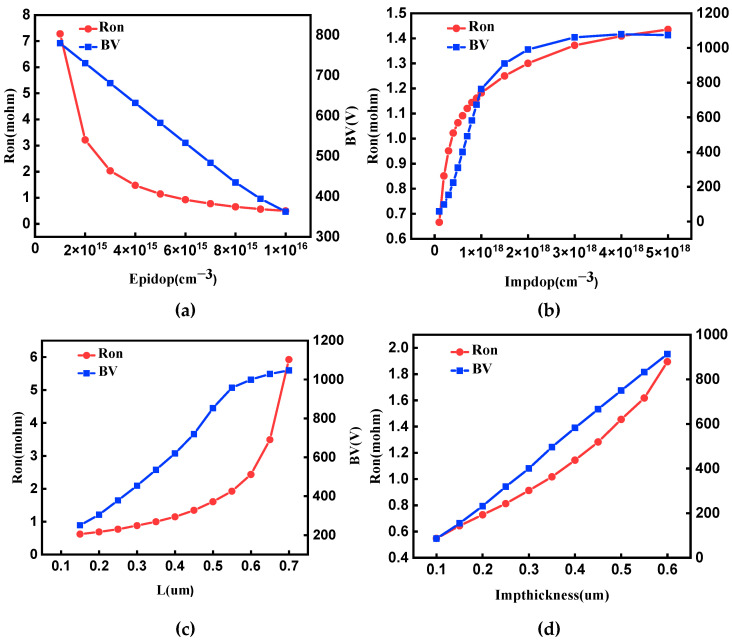
Ron and BV vary with different sensitive parameters, The corresponding sensitive parameters in Figure (**a**–**d**) are, respectively, doping concentration in drift region (Epidop), doping concentration in P+ region (Impdop), a ratio of the width of P+ region to the spacing between adjacent P+ region (defined by parameter L), and injection depth of P+ region (Impthickness).

**Figure 3 micromachines-14-00188-f003:**
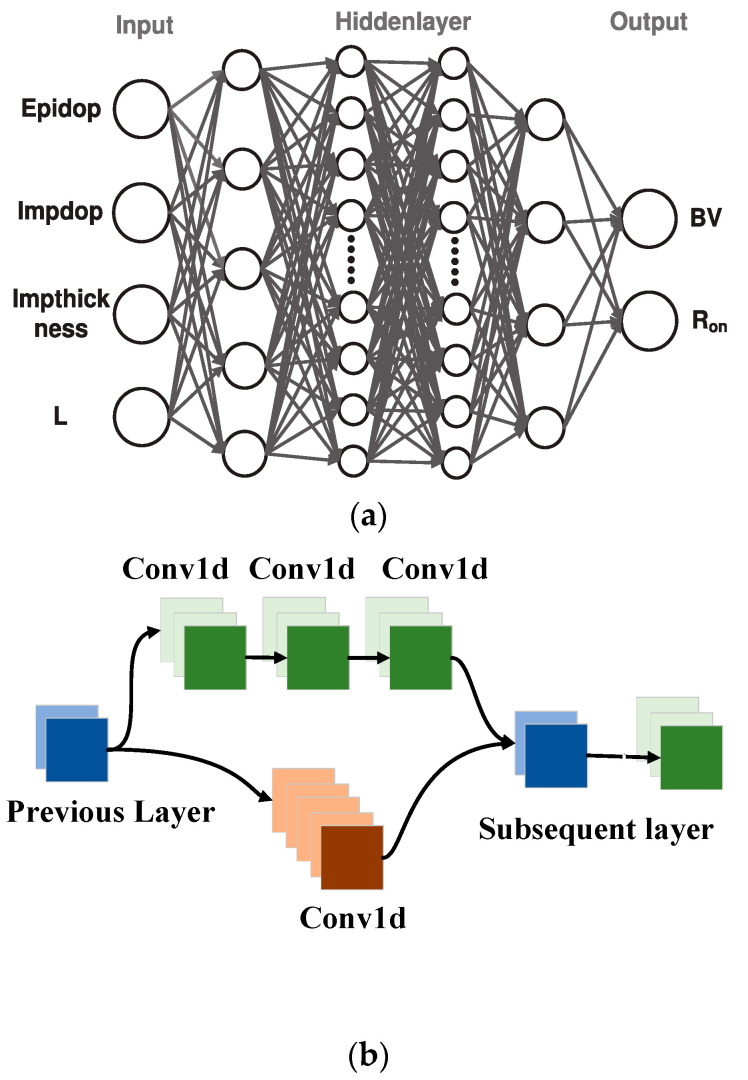
Neural network structure. (**a**) The basic neural network structure comprises input, hidden, and output layers. (**b**) Structural diagram of double-branch convolution module, one branch contains three successively cascaded convolution kernels of 3 × 1 one-dimensional convolution layer, and the other includes a convolution kernel of 5 × 1 one-dimensional convolution layer.

**Figure 4 micromachines-14-00188-f004:**
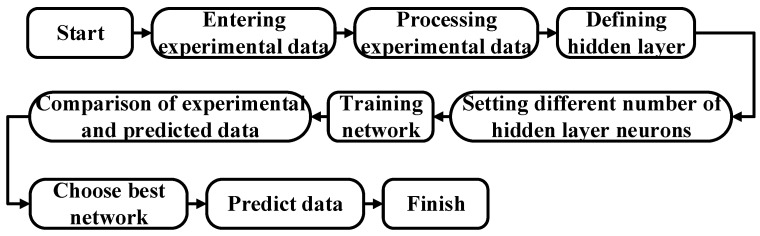
The flow chart for determining the optimal network Structure.

**Figure 5 micromachines-14-00188-f005:**
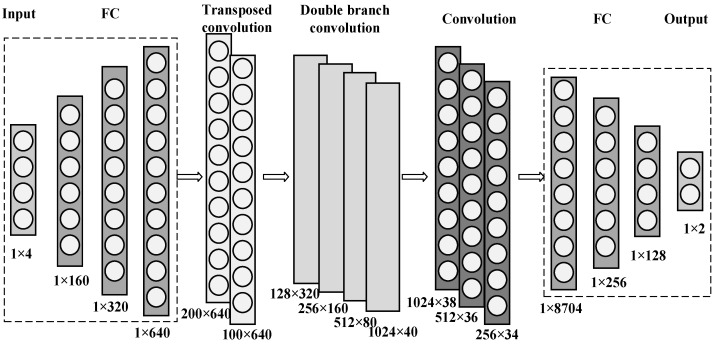
The neural network structure diagram finally determined in this paper includes, from left to right, four input layers, three fully connected modules, two transpose convolutional modules, four double-branch convolutional modules, three convolutional modules, three fully connected modules, and two output layers. At the bottom of each layer, the number of neurons is marked.

**Figure 6 micromachines-14-00188-f006:**
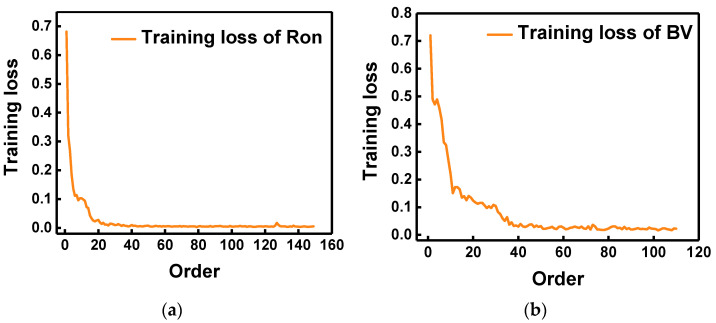
Training loss in the training process. (**a**,**b**) are training loss of Ron and BV, respectively. With the increase in the number of iterations, the loss in the training process gradually decreases and finally becomes stable.

**Figure 7 micromachines-14-00188-f007:**
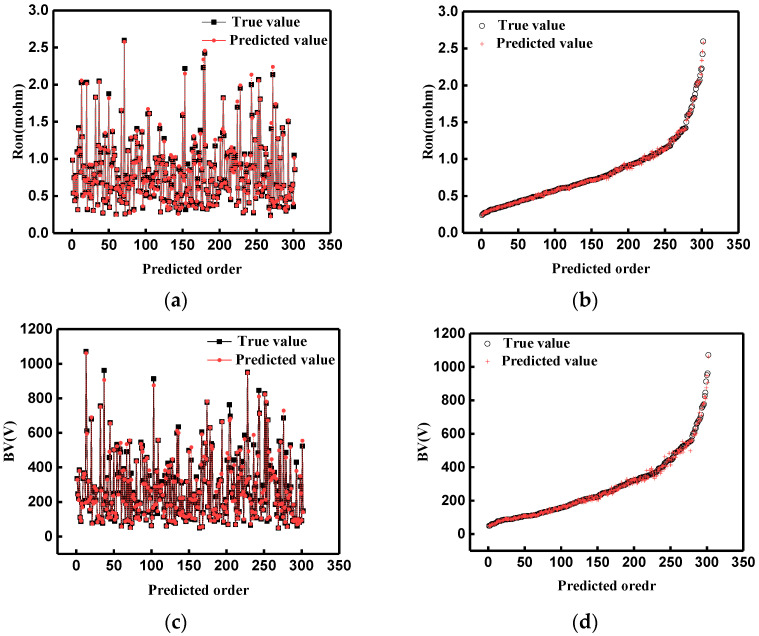
The prediction results of Ron and BV. (**a**) The disordered prediction results of Ron. (**b**) The prediction results of Ron under forward ranking. (**c**) The disordered prediction results of BV. (**d**) The prediction results of BV under forward order.

**Figure 8 micromachines-14-00188-f008:**
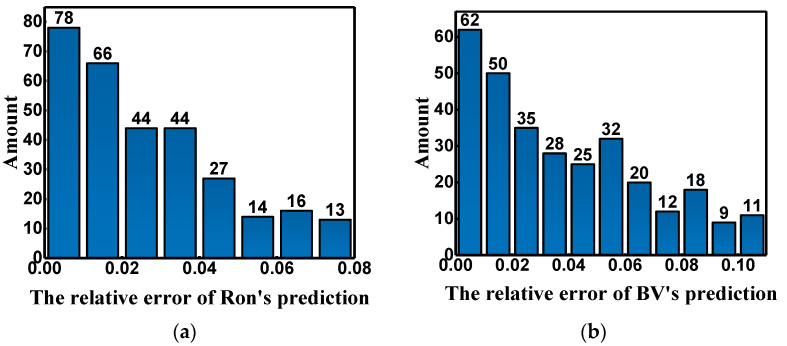
The error distribution histogram of the predicted results; (**a**,**b**) are the error distribution histograms of Ron and BV, respectively.

**Figure 9 micromachines-14-00188-f009:**
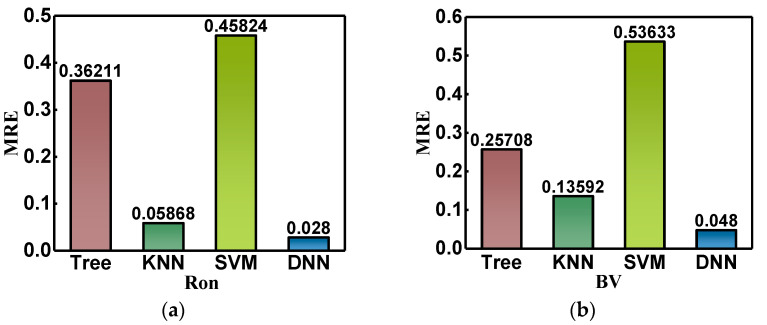
The mean relative errors of the predicted results of the neural network established in this paper are compared with those of traditional machine learning methods; (**a**,**b**) are the comparison of Ron and BV, respectively.

**Table 1 micromachines-14-00188-t001:** GaN JBS device structure parameters.

Paraments	Values
Total Width	10 µm
Drift Region Thickness	5 µm
Substrate Thickness	10 µm
P+ Region Thickness (Impthickness)	0.4 µm
P+ Region Width (2 L)	0.8 µm
P+ Gap Width (2−2 L)	1.2 µm
Drift Region Doping (EpiDop)	5 × 10^15^ cm^−3^
Substrate Doping	1 × 10^19^ cm^−3^
P+ Region Doping (Impdop)	1 × 10^18^ cm^−3^

**Table 2 micromachines-14-00188-t002:** The control group of each sensitive parameter in the sample data.

Paraments	Values
Drift Region Doping (cm^−3^)	3 × 10^15^, 4 × 10^15^, 5 × 10^15^, 6 × 10^15^, 7 × 10^15^, 8 × 10^15^, 9 × 10^15^, 1 × 10^16^
P+ Region Doping (cm^−3^)	3 × 10^17^, 4 × 10^17^, 5 × 10^17^, 6 × 10^17^, 7 × 10^17^, 8 × 10^17^, 9 × 10^17^, 1 × 10^18^
The ratio of P+ region width to adjacent P+ region spacing	2:8, 1:3, 3:7, 7:13, 2:3, 9:11, 1:1, 3:2
P+ Region Thickness (µm)	0.15, 0.20, 0.25, 0.30, 0.35, 0.40

## Data Availability

Not applicable.

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
