# Peer review of "GaN JBS Diode Device Performance Prediction Method Based on Neural Network"

_micromachines, 2023, doi:10.3390/mi14010188_

Round 1

Reviewer 1 Report

The manuscript reports the method about GaN JBS diode device performance prediction method based on neural network. There is a good fitting result between the predicted value of Ron and BV and the real value. The manuscript should be published in Micromachines after authors amend some minor items listed below.

1. The good fitting result between the predicted value of Ron and BV and the real value should be mentioned in abstract and conclusion.

2. The introduction does not introduce much of the previous research work . Whether there is any research on the application of neural networks on GaN JBS diode device, these studies should be compared with this manuscript.

3. In order to avoid confusion among readers, there should be text annotations in Figure 3b and Figure 5.

Reviewer 2 Report

This manuscript proposed a prediction method for predicting GaN JBS devices using ANN. The topic is interesting and timely, however, due to the lack of novelty and other flaws, the manuscript cannot be accepted in its current form – A major revision is required before re-consideration.

Major concerns:

1)     Using ANN to predict device performance is not a novel topic. There have been already many works addressing such problems, like 10.1109/TED.2021.3063213 and 10.48550/arXiv.2105.11453. To what extent do the authors reckon the proposed method is promising? It is suggested to give further emphasis, regarding the references mentioned above.

2)     The training/test/validation dataset should be introduced in detail, i. e., how to prepare the data (in encoding), how to wash data, and how to construct a database.

3)     The machine learning framework used here should be introduced. Also, the computer configuration should be introduced, i. e., which CPU/GPU etc.

4)     In Fig.1 pls label how many neurons are there in each layer?

5)     In Fig.4 is it just a single shot? If there are multiple processes, please label how to loop.

6)     Show training loss and accurate vs iteration data.

7)     The original dataset should be provided as, e. g., supplementary material for further check.

Minor concerns:

1)     The manuscript must be proofread – Many typos and dislocations.

2)     Give a discussion on the generalization of the model.

3)     Give a discussion on how to avoid the over-fitting issue in this task.

Overall, the presented manuscript’s topic is timely and interesting. But because the manuscript, in its current form, lacks some critical information, it cannot be recommended for publication. A major revision is required before reconsideration.

Round 2

Reviewer 2 Report

Thanks for the revision. The paper addressed all my concerns and now is ready to go.